# Human Exome Sequencing and Prospects for Predictive Medicine: Analysis of International Data and Own Experience

**DOI:** 10.3390/jpm13081236

**Published:** 2023-08-08

**Authors:** Oleg S. Glotov, Alexander N. Chernov, Andrey S. Glotov

**Affiliations:** 1Department of Genomic Medicine, D. O. Ott Research Institute of Obstetrics, Gynecology and Reproductology, 199034 St. Petersburg, Russia; anglotov@mail.ru; 2Department of Experimental Medical Virology, Molecular Genetics and Biobanking of Pediatric Research and Clinical Center for Infectious Diseases, 197022 St. Petersburg, Russia; 3Department of General Pathology and Pathological Physiology, Institute of Experimental Medicine, 197376 St. Petersburg, Russia

**Keywords:** whole-exome sequencing, personalized medicine, human monogenic diseases, oligogenic and multifactorial diseases

## Abstract

Today, whole-exome sequencing (WES) is used to conduct the massive screening of structural and regulatory genes in order to identify the allele frequencies of disease-associated polymorphisms in various populations and thus detect pathogenic genetic changes (mutations or polymorphisms) conducive to malfunctional protein sequences. With its extensive capabilities, exome sequencing today allows both the diagnosis of monogenic diseases (MDs) and the examination of seemingly healthy populations to reveal a wide range of potential risks prior to disease manifestation (in the future, exome sequencing may outpace costly and less informative genome sequencing to become the first-line examination technique). This review establishes the human genetic passport as a new WES-based clinical concept for the identification of new candidate genes, gene variants, and molecular mechanisms in the diagnosis, prediction, and treatment of monogenic, oligogenic, and multifactorial diseases. Various diseases are addressed to demonstrate the extensive potential of WES and consider its advantages as well as disadvantages. Thus, WES can become a general test with a broad spectrum pf applications, including opportunistic screening.

## 1. Introduction

In recent decades, scientific and technological advances in biology and medicine have produced new high-tech methods for early diagnostics and paved the way to the identification of genetic markers and the introduction of new screening strategies in clinical practice. This has enabled clinicians to precisely identify the causes of rare MDs, improve prevention, and boost the efficiency of treatment for multifactorial socially significant diseases, ultimately contributing to better health quality and life expectancy among the populations of economically developed countries [1]. All of these achievements have given an impetus to a paradigm shift in the overall healthcare system and enabled the transition from group-based to predictive or preventive personalized medicine (PM) and therapy that rely on a disease’s clinical diagnosis and stage, a patient’s gender and age, as well as the individual molecular genetic biomarker profiles associated with pathology development, prognosis, outcome, and treatment efficacy. Advances in genetics and information technology allowed for the emergence of new disciplines, such as genomics (proteomics, metabolomics, transcriptomics, and pharmacogenomics), and shaped the set of standard criteria for large dataset processing using bioinformatics, high-throughput techniques, and new-generation DNA sequencing (NGS) to study gene structure [1].

The identification of single-nucleotide variants (SNVs, SNPs) raised further challenges in annotating pathological variants and associating them with diseases. The HapMap and 1000 Genomes projects data were used to develop a methodology for genome-wide association studies (GWASs) [2]. GWASs allowed for the characterization of population frequencies of multiple SNVs/SNPs associated with multifactorial diseases (MFDs), such as type 1 diabetes (T1D) [3] and type 2 diabetes (T2D) [4].

In 2009, Shendure’s research group pioneered the use of whole-exome NGS to detect genetic aberrations and eventually discovered Miller–Fisher syndrome, a rare recessive inflammatory (autoimmune) demyelinating polyradiculoneuropathy [5]. Since 2012, it has become clear that as soon as we understand the type of inheritance, disease pathogenesis, and SNV/SNP population frequencies, the annotation and clinical interpretation of NGS-identified pathogenic gene variants are no longer impossible [6]. Notably, the contemporary identification of genes and their variants associated with a particular disease (condition) enables us to study their pathogenetic role.

NGS applications in medical research are versatile. Via their intended use, high-throughput sequencing techniques can be split into the following groups: (1) analysis of entire genome (whole-genome sequencing, WGS); (2) analysis of protein-coding genes in a genome (whole-exome sequencing, WES); (3) analysis of particular disease-causing gene sequences (from clinical exomes embracing some 4000–5000 clinically relevant genes, to kits for small target regions of one–three genes or loci); (4) transcriptome sequencing (RNA-seq); and (5) analysis of bacterial microbiome biological diversity, Table 1 [1,7].

Considering that the purpose is to limit the examination to protein-coding nucleotide sequences to identify rare pathological SNPs, insertions, deletions that may underlie a disease, or the identification of new genetic markers of oligogenic diseases and MFDs, WES is a more cost-efficient technique than WGS [8]. Versatile investigations show that the performance of exome tests is only 2% inferior than that of WGS tests, which detect 44 to 50% of all pursued mutations [9,10]. This is largely conditioned by the fact that exome analyses utilize longer fragments and more advanced probe designs [11]. WES occupies a special place among NGS tests. Considering its relatively low cost, WES is currently more attractive in clinical settings. Furthermore, WES allows for the significant minimization of the size of the analyzed database to 6 GB, in contrast to WGS (90 GB) [12]. For example, WES is 10-fold more efficient in diagnosing MODY mutations [13]. In addition, WES is becoming more and more famous, thanks to various tools for its analysis. For instance, in combination with a unique bioinformatic approach, this method enabled us to identify over 10 new markers for T2D [14,15,16].

Today, WES is becoming a first-line examination procedure for sophisticated investigatory as well as practical purposes. WES/WGS allows for the identification of new unknown or rare variants that are associated with diseases or very rare pathologies, or are present in some populations, but have been overlooked in differential diagnoses so far without being considered to cause disease. Sherri Bale et al. utilized WES to diagnose disorders of the central nervous system (31%, N = 1082), multiple congenital anomalies (36%, N = 729), and cardiovascular (28%, N = 54), skeletal (39%, N = 54), musculoskeletal (40%, N = 43), vision (47%, N = 60), skin (32%, N = 31), and hearing (55%, N = 11) abnormalities in 3040 patients, with a WES total yield of 28.8%. These included 2091 cases, where 6.2% (N = 129) had pathogenic variants [17]. Using WES, Suyun Qian et al. diagnosed metabolic, neuromuscular, and multiple deformities (MDs) in 25% (43 of 169) of critically ill children [18]. In people with severe developmental delays, microcephaly, seizures, dysmorphic facial features, and poor muscle mass, 29,860 variants in 19,160 genes have been studied via the use of WES. A pathogenic variant was detected in the *ASXL* transcriptional regulator-3 (*ASXL3*) gene, which causes Bainbridge–Ropers syndrome [19].

A meta-analysis published in 2023, including 50,417 probands, provides similar diagnostic rates for WES and WGS and shows the greater clinical utility of WGS compared to WES. With the recent downward trend in the cost of VUS found in noncoding regions of the genome and WGS, it is expected that WGS will be widely used in clinical research [20].

However, not all populations have been well studied so far. Recent years have seen the publication of numerous studies demonstrating a variety of population characteristics for allele frequencies of functionally significant polymorphisms. It is therefore apparent that each population has its own spectrum of SNPs. It is thus critical to investigate the population frequencies of functionally significant gene alleles in early stages of research. The importance of population studies is explained through frequency differences in functional polymorphic gene variants across different populations, which may depend on geographical conditions, regions of residence, dietary features, race and ethnicity, and many other factors [21,22]. Therefore, there is little doubt that, in different populations or ethnic groups, the way that identical genetic polymorphisms affect the etiology and pathogenesis of a particular disease is never the same [22,23,24].

WES, like any other technology, shows a set of drawbacks due its technical characteristics. Thus, the processing and analysis of a large amount of WES data becomes a bottleneck, making it difficult to differentiate small mutations from random errors that occur during sequencing [25]. In addition, the main disadvantage of WES is the uneven coverage of DNA code reads over target genes, which results in many low-coverage regions that prevent the accurate annotation and interpretation of variants [26]. WES data may include inconsistencies, such as outliers and anomalies, or inconsistent speeds at which data are loaded onto the repository, GC bias, the association of variants with biological traits, and phenotype. The interpretation of sequencing results for clinical diagnosis is another limitation [8]. Therefore, all of this requires the clear systematization of existing knowledge to establish the concept of the application of WES in clinical practice.

The purpose of our review is to study the risk factors for socially significant diseases based on WES results and to develop methods allowing for the identification of clinically relevant gene variants in order to assess the risk of monogenic, oligogenic, and multifactorial pathologies.

## 2. Human Monogenic Diseases: Population Genetics Research

### 2.1. Human Monogenic Diseases

The OMIM database (as of 1 June 2023) includes entries for 7377 hereditary diseases and syndromes, as well as their molecular associations [27]. These include 6305 phenotypes associated with one single gene, i.e., showing the monogenic nature of a genetic trait or syndrome. This was largely achieved due to the active implementation of WES and the exome consortium [28,29,30].

### 2.2. Population Genetic Researches for Monogenic Diseases

The genetic structure of human populations has been extensively studied worldwide. Nonreference (i.e., non-wild type) allele frequency in s particular population is a most important factor influencing the clinical interpretation of a genetic variant. Genetic variability in many regions of the world is poorly understood despite the very large number of variants (125,748) in the genome aggregation database (gnomAD, version v. 2.1). Wenhao Zhou et al. analyzed the prevalence of cystic fibrosis (CF) using 30,951 WES (20,909 pediatric and 10,042 parent) samples and compared these with those of Caucasians [31]. After filtration, 477 variants of the cystic fibrosis transmembrane regulator (*CFTR*) gene were left, and 53 variants were annotated as pathogenic/probably pathogenic (P/LP). The authors used the annotated variants to evaluate the prevalence of CF in China to be 1/128,434. Only 39.6% (21/53) of the variants were used to screen for CF in Caucasians, producing underestimated values for the prevalence of CF in China among children (1/143,171 vs. 1/1,387,395, *p*  =  5 × 10^−24^) and an adult population (1/110,127 versus 1/872,437, *p*  =  7 × 10^−10^). The allele frequencies of six (L88X, M469V, G622D, G970D, D979A, and 1898+5G->T) pathogenic variants were higher in a Chinese population compared with a gnomAD non-Finland European population (all *p* <  0.1). Using haplotype analysis, the researchers showed greater diversity in haplotypes in a Chinese population compared to Caucasians. The founder mutations of the Chinese and Caucasians were G970D and F508del, with two SNPs (rs213950–rs1042077) identified as related genotypes in an exon region.

Our investigations did not identify prevalent pathogenic SNPs missing from ClinVar or dbSNP in autosomal recessive disease-causing genes. This indicates that the majority of disease alleles are common for Russian and European populations, at least for disorders with recessive inheritance patterns. These results allowed us to suggest preliminary estimates for the prevalence of monogenic disorders, based on the identified exome variants for the region (Table 2).

Although the small sample size does not allow us to reliably determine the extent of discordance, our results for CF and phenylketonuria are consistent with estimates for these genes [32]. Remarkably, our findings show that, in Northwest Russia, Stargardt disease is more prevalent than cystic fibrosis, as has been the belief [32].

The research also looks into pathogenic variants for a number of human diseases. The results are formulated in Table 1, showing the diseases with the highest prevalence: Stargardt disease caused by mutations in the *ABCA4* (MIM#601691) gene, which has also been previously reported [33]. Our results are concordant with earlier large-scale research into the incidences of pathogenic alleles associated with cystic fibrosis in a non-Finnish European population [42]. Our estimates of CF, phenylketonuria, and galactosemia prevalence were concordant with those of other genetic studies [32,43].

Thus, our results indicate the need to create genetic population databases for the interpretation of variants and the identification of disease risk factors.

### 2.3. WES Application to Identify New Variants in the Genomes of Patients

WES allows for the identification of new gene variants in patients with MDs. Doctors Daniel Trujillano, Rami Abou Jamra, et al., using WES, sequenced 2819 samples of 1000 patients from 54 countries with a wide phenotypic spectrum. Overall, they determined 320 pathogenic (P) or likely pathogenic (LP) and 303 unique variants from 1000 patients undergoing clinical WES, 307 (30.7%) of which had a positive gene finding. In addition, other findings included ethylmalonic encephalopathy (ETHE1), Niemann–Pick disease type C2 (NPC2), Temtamy syndrome, pyruvate dehydrogenase E1-alpha deficiency (PDHA1), galactosemia (GALT), propionic acidemia (PCCA), homocystinuria (CBS), CF, long QT syndrome, and polycystic kidney disease. This justifies the idea that highly heterogeneous pathologies can be effectively detected using WES. Among other findings, new genes were detected, such as non-receptor protein tyrosine phosphatase type 23 (*PTPN23*) associated with brain developmental delay and atrophy, potassium channel tetramerization domain containing 3 (*KCTD3*) causing severe intellectual disability and seizures, alpha three subunit of sodium voltage-gated channel (*SCN3A*) associated with autosomal dominant encephalopathy, protoporphyrinogen oxidase (*PPOX*) causing variegate porphyria and developmental delay, and FERM and PDZ domain-containing 4 protein (*FRMPD4*) implicated in X-linked intellectual disability as well as recessive Dravet syndrome. The total WES diagnostic rate stands at 31% [44]. In another study, Joanne Trinh et al. sequenced 26,119 exome samples from 4351 patients with neurodevelopmental disorders (NDDs), such as global developmental and motor delay, macrocephaly, microcephaly, seizures, and delayed speech and language development. Researchers determined 65 rare variants in 14 genes. The 14 detected variants were classified as P or LP and included cyclin dependent kinase 13 (*CDK13*), chromodomain helicase DNA binding protein 4 (*CHD4*), potassium voltage-gated channel subfamily Q member 3 (*KCNQ3*), lysine methyltransferase 5B (*KMT5B*), transcription factor 20 (*TCF20*), and C2H2-type zinc finger protein (*ZBTB18*). The 51 remaining variants (78%) belonged to the VUS category. Two of the patients had multiple molecular diagnoses, including P/LP variants in forkhead box G1 transcription factor (FOXG1), CDK13, and the transmembrane protein 237 (*TMEM237)* and *KMT5B* genes. The total WES diagnostic rate was 31% [45]. Zhang Q et al. sequenced 1360 patients to identify 604 genetic pathologies associated with 150 genetic syndromes, 510 genes, and 718 variants. In this cohort, the overall WES positive identification rate for disease-related gene alteration was 44.41%. Investigators detected growth abnormalities in 49.37% (118/239), seizures in 44.54% (102/229), autism spectrum disorder in 32.76% (38/116), global developmental delay in 54.84% (51/93), motor deterioration in 48.06% (99/206), abnormalities of the respiratory system in 40.61% (67/165), cerebral palsy in 41.26% (59/143), and abnormalities of the head or neck in 55.52% (161/290), the skin in 53.70 (58/108), the endocrine system in 49.78 (112/225), hearing or vision in 58.51% (55/94), the skeletal system in 53.95% (116/215), and the cardiovascular system in 43.20% (54/125) of samples [46].

WES allows for the identification of new, very different variants in various populations. WES enabled us to identify new variants in the low-density lipoprotein receptor (*LDLR)* gene in 59 Russian patients with a history of familial hypercholesterolemia (FH) [47]. FH results from genetic variants in the *LDLR*, apolipoprotein B (*APOB*), and subtilisin/kexin proprotein convertase type 9 (*PCSK9*) genes [48]. FH-associated variants were determined in 25 children and 18 adults, showing mutation detection rates of 89 and 58% for the children and adults, respectively. In the adults, 13 patients had variants in the *LDLR* gene, 3 patients had *APOB* variants, and 2 had ATP-binding cassette transporter 5 (ABCG5)/G8 mutations. Twenty-one children had FH-associated variants in the *LDLR* gene; see Table 2. Our study identified seven novel pathogenic or likely pathogenic *LDLR* variants (Table 3). Among them, four missense variants were located in the protein coding regions, and two were frameshift mutations responsible for the production of truncated proteins. These mutations were only reported in one patient, whereas an intron 6 splicing variant (c.940+1_c.940+4delGTGA) was detected in four unrelated individuals. Variant p.Gly592Glu in the *LDLR* gene was identified in six (10%) Russian patients and may presumably constitute the main FH variant in the Russian population.

FH is a common, underdiagnosed, and untreated genetic disease worldwide [50]. Therefore, WES sequencing data can be used to detect new candidate genes.

Current sequencing methods allow for the detection of a bundle of hereditary diseases in an individual, thus gaining unprecedented significance. Such cases are not as rare as they may seem. For instance, we would like to refer to a case of the coinheritance of X-linked and dominant forms of ichthyosis [51]. This information may be valuable for genetic counseling because of similar clinical symptoms. It is therefore necessary to analyze both steroid sulfatase (STS) and filaggrin (FLG) genes to exclude combined forms of ichthyosis. Notably, NGS allows us to identify P or LP SNPs in genes that were earlier believed to possess mutations of a single type [52].

For a set of disorders, adequate therapy is the most critical outcome of NGS examination. In male probands with delayed growth and bone age, intellectual impairment, skeletal and facial features, and partial responses to hormone treatment, we identified a c.7466C>G (p.Ser2489*) heterozygous pathogenic mutation in the last exon of the SRCAP (Snf2 related CREBBP activator protein) gene, thus suggesting a new model of floating harbor syndrome (FHS) pathogenesis. These genetic mutations have dominant-negative effects that explain the limited efficacy of growth hormone treatment in FHS [53].

### 2.4. General Strategy and Algorithm of WES Implementation in Human Genetic Pathology Diagnostics

WES provides a robust technique for MD diagnosis in humans. Yingchao Liu et al. utilized WES to study 169 children with critical disorders (median age = 10.5 months) and MDs [18]. Monogenic disorders were diagnosed in 43 (25%) patients. Pathologies with the highest incidences included metabolic (33%) and neuromuscular (19%) diseases, as well as multiple deformities (14%). The efficacy of diagnoses in children with metabolic disorders, growth impairment, or ocular abnormalities improved once thorough clinical data were available. WES data enabled adjustments in 30 (70%) cases, including disease monitoring initiation in 41.9% (18 cases), rehabilitation and palliative care in 27.9% (12 cases), the modification of ongoing treatment in 25.6% (11 cases), other comprehensive evaluation procedures in 7% (3 cases), and family intervention in 4.7% (2 cases).

Tasja Scholz et al. studied the diagnostic efficacy of WES for MDs to identify phenotypes in 61 infants with critical idiopathic disorders [54]. Investigators performed one single WES, two duo-WES, and fifty-nine trio-WES. The overall diagnostic rate was 46% (28/61) and 50% (15/30) in neonate subgroups. The yielded data showed that WES is a noninvasive diagnostic tool with a high rate of MD identification in neonates and infants. Thus, the evidence justifies the application of WES as a first-line examination for preconception genetic diagnosis and in idiopathic disorders in probands with a “blurred” phenotype.

To ensure efficiency (see Table 4), the following cost-effective strategy is suggested for the genetic diagnosis of MODY, WD, and other MDs associated with major mutations. We also show additional benefits of the application of WES in disease diagnosis.

The obtained data are concordant with the global assumptions (7.7 × 10^−6^) [31].

It should be noted that NGS does not always suffice to formulate a diagnosis; hence, in some cases, concurrent or subsequent Sanger sequencing is required to detect the other pathogenic variant. In patients with a blurred clinical picture, differential diagnosis with WES is necessary to identify the root cause of a disease. For example, NGS was used to analyze the hotspot region in the RNA processing endoribonuclease (*RMRP)* gene promoter in a proband with extremely rare autosomal recessive skeletal chondrodysplasia (anauxetic dysplasia, AD). Heterozygous rs387906533 (n.91_92delinsGC) variants of the nucleotide sequence (chr9:35657924-35657925delCTinsGC) were detected in exon 1 of the RMRP gene and an unknown n.–6_–5insTCTCAGCTTCAC substitution (chr9:g.35658020 35658021insTCTCAGCTTCAC) in the gene promoter region; see Figure 1. The variant is a 12-nucleotide insertion between the TATA box and the transcription start site [61].

It was found that the n.–6_–5insTCTCTCAGCTTCAC mutation was of paternal origin and the n.91_92delinsGC mutation was of maternal origin. No prior evidence has ever been reported regarding the insertion in the *RMRP* gene promoter region as a cause of AD with no extraskeletal manifestations (typical of carriers of similar mutations) [61].

## 3. New-Generation Sequencing, Phenotypic Screening, Oligogenic and Multifactorial Diseases

Hereditary diseases caused by pathogenic variants in a few genes are much more prevalent—i.e., the so-called oligogenic hereditary diseases [63]. Oligogenic diseases are an interim condition between MDs, associated with one specific defected gene, and polygenic diseases, caused by several genes and exogenic factors.

### 3.1. Oligogenic Etiology of Cardiomyopathies

Understanding the disease origin is therefore critical. Whether the disorder is mono- or oligogenic, or multifactorial, the answer is often far from evident. WES is a promise to answer this question. Pak-Chung Sham et al. utilized WES to identify six new P or LP variants in 40 patients with hypertrophic (HCM, n = 14) and dilated cardiomyopathy (DCM, n = 26) [63]. Hypertrophic cardiomyopathy caused by gene variants coding sarcomeric proteins—β-myosin heavy chain (MYH7) and myosin binding protein C (MYBPC3)—account for up to 50% of all clinical cases: myosin light chain 2 (MYL2), myosin light chain 3 (MYL3), and cardiac troponin T (TNNT2) in 5–10%, cardiac troponin I (TNNI3) in 5%, cardiac troponin C (TNNC1) in < 1%, cardiac α-actin (ACTC1) in < 1%, α-tropomyosin (TPM1) in 1.5%, and cysteine and glycine rich protein 3 (CSRP3) [64]. Authors established that frameshift (11:47372858, c.A224insG+) mutations in the MYBPC3 gene and missense (rs193922390, c.5135 G>A, p.R1712Q) mutations in the MYH7 gene were pathogenic variants for HCM. Missense variants, such as rs138049878 (c.2608 C>T, p.R870C), rs727503260 (c.2302 G>C, p.G768R), and rs397516088 (c.1063 G>A, p.A355T) in the MYH7 gene, and rs199476306 (c.188 C>T, p.A63V) in the TPM1 gene, were rated as likely pathogenic for HCM. The diagnostic WES yield for HCM produced 43% (six variants from fourteen patients). The missense variant rs121964856 (c.260 G>A, p.R87Q) in the TNNT2 gene was evaluated as pathogenic for DCM. The frameshift 2:179423322 (c.A60242del, p.S20082) and splicing 2:179549632 (c.13859) variants in the TTN gene were likely pathogenic for DCM. The diagnostic WES yield for DCM produced 12% (three variants from twenty-six patients).

Our study provides a follow-up to the research carried out by fellow investigators, with the aim of shedding light on the mode of inheritance of cardiomyopathies and to identify potential risk markers of the disease. The study subjects therefore include patients and conditionally healthy donors with different (favorable or unfavorable) histories [65,66,67]. According to the SNP SIFT analysis, substitutions in the TNNT2 gene were the most remarkable variants (Table 5).

We therefore suggest that HCM is a polygenic, rather than a monogenic, disease.

### 3.2. Monogenic Diabetes Mellitus

Monogenic maturity onset diabetes of the young (MODY) is yet another disorder with a multifactorial etiology, which constitutes 1–6% of diabetes mellitus (DM) cases in children and adolescents [69]. To date, 13 types of MODY are known, associated with the expression of 13 genes causing moderate or manifest hyperglycemia [70]. Due to the wide variety of clinical forms induced by numerous MODY-associated gene variants, various treatment approaches are used, from diet and physical activity to insulin therapy.

Zanchao Liu et al. applied WES and Sanger sequencing to analyze genetic markers in 200 MODY patients from Northern China [71]. The researchers found a rare rs535471991 (c. T1895G) mutation in the phosphorylated domain of the forkhead box M1 transcription factor (FOXM1) gene, which impairs the functionality and proliferation of pancreatic β-cells. These results suggest that the FOXM1 gene is involved in the pathogenesis of MODY. These findings can pave the way towards future target therapy for MODY patients.

Philippe Froguel et al. performed WES for the identification of mutations in three affected and one healthy relative in the MODY-X family, compared to 406 controls [72]. In total, 324 variants were detected in the study family and controls. It was c.679G>A (p.Glu227Lys) substitution in potassium inwardly rectifying channel subfamily J member 11 (KCNJ11), however, that was responsible for MODY-X in the study family (LOD score of 3.68).

Today, WES enables investigators to identify pathogenic variants in other non-glucokinase (GCK) genes [13]. Three patients were identified as having different variants of target genes. Patient 1 had a GCK in-frame deletion that was associated with a hepatocyte nuclear factor-1 alpha (HNF1A) missense mutation (patient #226). Patient 2 had two missense substitutions in GCK and Src family tyrosine kinase proto-oncogene (BLK, patient #529). Patient 3 (#662) reported splicing in GCK and a missense mutation in BLK in the presence of wolframin ER transmembrane glycoprotein (WFS1) variants. Clinical symptoms in patient #226 were more typical for MODY2. Both patients #529 and #662 had clinical manifestations of GCK-MODY. Our data show the absence of severe pathogenic effects caused by detected non-GCK variants [13]. Figure 2 shows the spectrum of P variants associated with neonatal diabetes mellitus (NDM) and MODY.

Pathogenic variants in the GCK gene can be associated with different clinical disease manifestations [13].

Monogenic diabetes comprises an amount of non-MODY transient forms associated with 20 genes [73]. Neonatal diabetes can be inherited in a dominant or recessive manner and manifests as various syndromes [74]. Hyperglycemia is often diagnosed prior to the manifestation of other symptoms due to the extremely early onset of diabetes. The treatment approaches for neonatal diabetes without MODY are based on the presence of a specific genetic mutation that causes the diabetic phenotype. Our data regarding pathogenic variants in a single gene and their link with various diseases or DM manifestations (MODY and NDM) emphasize the importance and cautious implementation of “penetrance” and “expressiveness” as concepts describing such genetic syndromes in clinical practice. Nosology varieties could also originate from different variants of transformed function in the same genes, exacerbated by environmental factors.

### 3.3. Multifactorial Diseases (MFDs): Type 2 Diabetes Mellitus

Multifactorial diseases (MFDs) include almost all most prevalent chronic disorders, i.e., atherosclerosis, diabetes, obesity, bronchial asthma, osteoporosis, endometriosis, malignant tumors, and neuropsychiatric as well as cardiovascular diseases, arising from the interaction of many genes with adverse environmental factors [75]. Currently, the International Classification of Diseases (ICD) includes more than 55,000 nosological units [76]. The vast majority of them belong to MFDs. As of 11 March 2023, over 12,000 human diseases, 220,322 SNPs, and 493,105 genome associations were registered [77]. Today, there are three main approaches to the identification of candidate genes: the functional mapping method (candidate gene analysis), genetic linkage in high-risk families, and GWASs, including genome sequencing. GWASs are actively used to analyze and test samples from different national biobanks, including the UK Biobank [78].

According to the current level of knowledge, the influence of genetic factors on the expression and penetrance of phenotypic traits is due to the presence of genetic polymorphisms of point mutations with strong effects, on the one hand, or frequent SNPs with weak effects, on the other hand [79]. To understand the genotype–phenotype correlation, the preferable approach is to investigate candidate genes associated with MFDs [80].

Insulin-resistant T2D and obesity are very frequent chronic pathologies with multifactorial etiology with 128 and more than 700 genetic markers, respectively [81,82]. The genetic framework of T2DM and obesity has been studied using GWASs [65]. In this study, Mark McCarthy, using the 1000 Genomes multiethnic reference panel, conducted a GWAS analysis on 26,676 T2D patients and 132,532 control European individuals to detect 13 novel T2D-associated loci (*p* < 5 × 10^−8^) near the glucagon like peptide 2 receptor (*GLP2R*), gastric inhibitory polypeptide (*GIP*), and human leukocyte antigen (*HLA-DQA1*) genes [82]. SNVs included the following: rs60780116 (4:185708807, T>C, *p* = 7.38 × 10^−8^) in the long-chain fatty-acid-coenzyme A ligase (*ACSL1*) gene, rs2292626 (10:124186714, C>T, *p* = 1.75 × 10^−12^) in pleckstrin homology domain containing A1 (*PLEKHA1*), rs1061810 (11:43877934, A>C, *p* = 5.29 × 10^−9^) in the hydroxysteroid 17-beta dehydrogenase 12 (*HSD17B12*) gene, rs2925979 (16:81534790,T>C, *p* = 2.72 × 10^−8^) in the C-Maf inducing protein (*CMIP*) gene, rs78761021 (17:9780387, G>A, *p* = 5.49 × 10^−8^) in the glucagon like peptide 2 receptor (*GLP2R*) gene, and rs79349575 (17:46967038, A>T, *p* = 2.61 × 10^−7^) in the *GIP* gene. In general, 128 SNVs at 113 loci were associated with T2D.

In another study, John Chambers et al. carried out a GWAS analysis on 16,677 T2D patients from South Asian and 33,856 controls, and observed 21 new genetic loci for T2D with a significant association (*p*  = 4.7  × 10^−8^–5.2  ×  10^−12^) [83]. Among them, the most statistical significances with T2D were as follows: rs10916784 (1: 20729451, G, *p* = 5.2 × 10^−12^) in the von Willebrand factor A domain containing 5B1 (*VWA5B1*) gene, rs74790763 (5:122675214, C, *p* = 3.2 × 10^−11^) in the centrosomal protein 120 (*CEP120*) gene, rs62486442 (8:12623463, A, *p* = 2.4 × 10^−10^) in the LON peptidase N-terminal domain and ring finger 1 protein (*LONRF1*) gene, rs13257283 (8:105608497, G, *p* = 4.2 × 10^−8^) in the LDL receptor related protein 12 (*LRP12*) gene, and rs9568861 (13:54079446, T, *p* = 2.5 × 10^−8^) in the olfactomedin 4 (*OLFM4*) gene.

Our investigation displays that WES can serve as a technique to detect novel genes, associated with T2D and obesity [48]. The research included the detection of novel SNPs and loci for T2D and obesity in 110 Russian patients based on biologically meaningful filtering criteria. We have determined SNPs that serve as markers for T2D (rs1126930, rs9379084), obesity (rs11960429), and body mass index (rs1956549, rs11553746, and rs7195386). Using this approach, we detected rs11863726 in hemoglobin subunit theta 1 (*HBQ1*), rs328 in lipoprotein lipase (*LPL*), and rs112984085 in Vav guanine nucleotide exchange factor 3 (*VAV3*) for T2D and obesity. We also identified rs6271 in dopamine beta-hydroxylase (*DBH*), rs34042554 in protocadherin alpha 1 (*PCDHA1*), rs144183813 in pleckstrin homology domain containing A5 (*PLEKHA5*), rs62618693 in glutamine and serine rich 1 (*QSER1*), and rs61758785 in RAD51 paralog B (*RAD51B*) for obesity, in addition to rs685523 in ADAM metallopeptidase with thrombospondin type 1 motif 13 (*ADAMTS13*), rs2233984 in chromosome 6 open reading frame 15 (*C6ORF15*), rs17801742 in collagen type II alpha 1 chain (*COL2A1*), rs61737764 in integrin subunit beta 6 *(ITGB6*), and rs9379084 in Ras responsive element binding protein 1 (*RREB1*) for T2D in Russian patients [14,84,85,86].

Notably, in our investigation we employed a multiperspective approach (Figure 2) to detect candidate SNPs of T2D and obesity in a Russian population, which turned out to be a practical approach for limited-cohort studies (Figure 3). We used SNP and gene association tests based on the filtering of protein-altering SNPs and the prioritization of case- or control-specific genetic substitutions. We discovered that this approach prioritizes SNPs of middle and low frequency with a higher OR.

In conclusion, our research displays that WES is a rational approach that allows for the identification of MFD-associated genetic markers in limited populations. Thus, this strategy facilitates the identification of disease genes for polygenic traits.

### 3.4. Prospects of Comprehensive Individualized Screening for MFD Polygenic Factors

Although the polygenic inheritance of most common SNPs may exert certain mild underlying effects contributing to most common pathologies, the main effects belong to rare and common variants’ interactions [87]. It is currently unknown as to whether the application of a genome-wide polygenic score (GPS) to polygenic predictors would enable the detection of individuals at an increased clinical risk on a level comparable to rare monogenic substitutions [88].

Earlier attempts to develop such a GPS had limited success due to insufficient risk stratification for clinical utility. These efforts were hindered by three problems: (i) the small size of GWASs; (ii) computational limitations; and (iii) the lack of big datasets required for GPS validation. Khera and colleagues studied whether a GPS can detect groups of individuals bearing a level of risk exceeding that of monogenic variants. They analyzed five frequent pathologies: breast cancer, T2D, CAD, atrial fibrillation, and inflammatory bowel disease. For each of the illnesses, a GPS was developed based on extensive GWAS data from European participants and the UK Biobank genotype data. The predictors showed an AUC of between 0.79 and 0.81 in the validation dataset, with the best predictor (GPSCAD) involving 6,630,150 mutations. This predictor was well manifested in the test cohort with an AUC of 0.81 (Table 6).

The advantage of GPSCAD is that it can be assessed starting from birth, well before the discriminative capacity of risk factors used in clinical practice to predict CAD. Information on high GPSCAD values and individual hereditary predisposition may facilitate prevention efforts. For example, we showed that a high CAD risk could be offset by a healthy lifestyle or cholesterol-lowering treatment with statins [88]. Similar results were obtained for four other conditions. T2D is a main factor of cardiovascular and renal pathologies. The polygenic predictor determined 3.5% of the population as having a 3-fold risk and the top 1% as having a 3.30-fold risk [88]. Both drugs and cardinal lifestyle interventions prevent progression to T2D [89]. Polygenic scores make a quantitative metric of personal hereditary risk based on the cumulative effect of multiple common SNPs. Risk prediction accuracy shall improve considerably with the advancement of exome and whole-genome data. There is no doubt that, within a few years, the combination of genome-wide sequencing data and genome-wide searches for allelic associations for all major MFDs is bound to dramatically increase the predictive value of presymptomatic hereditary predisposition testing, considering the rapid progress in molecular medicine [1].

However, WES cannot achieve a 100% diagnostic rate. Therefore, assistance will need to be sought from undiagnosed disease networks or collaborating with researchers to further characterize VUS variants.

### 3.5. Clinical Genetic Passport (CGP)

WES enables investigators to study the structure, genetic polymorphisms, and functions of different variants in the genome within specific populations, providing insights into the hereditary mechanisms of a particular monogenic disease or MFD and contributing to their diagnosis, prevention, and treatment, in anticipation of the future revised classification of human diseases. This, in turn, leads to a paradigm shift and propels the advent, as well as rapid development, of molecular predictive medicine and genetic passport frameworks as steppingstone towards a clinical genetic passport (CGP), resulting from exome sequencing data [90,91]. A comprehensive approach, based on an entire array of molecular, genetic, cytogenetic, and embryological methods, in pregnancy planning is desperately demanded today. CGPs, genetic mapping, and NGS are present in all areas of medical science, enabling clinical medicine to solve reproductive problems, among other issues [1,23], such as the following: noninvasive prenatal testing (NIPT) or screening for monogenic and oligogenic diseases via the detection of pathogenic variants in probands and high-risk families; pregnancy planning via the use of preimplantation genetic diagnostics (PGDs) and treatment; and diagnosis confirmation, prospectively extending into MFD and infectious (COVID-19) disease risk assessment in addition to the identification of phenotypic traits in humans [92,93,94] (Figure 4).

The above being the case, efficient CGP implementation requires ample population databases (including on a national level), reinforced by advanced bioinformatic and statistical protocols for sequencing data processing and analysis, in order to evaluate the incidence of SNPs associated with hereditary and other pathologies.

It is important to note that many genetic substitutions previously identified as pathogenic occur increasingly frequently in healthy persons, causing Mendelian inheritance diseases. This was the strongest factor in reducing false-positive associations of variants with phenotypes [95]. Of note are critically important updates in genetic terminology regarding the newly emerged methods. Instead of the common terms “mutation” and “polymorphism”, in 2015 and 2017 the American College of Medical Genetics and Genomics (ACMG) and the Russian Society of Medical Genetics, respectively, recommended the use of the term “nucleotide sequence variant” with the following modifiers: (1) P, (2) LP, (3) uncertain significance, (4) likely benign, or (5) benign [96,97]. Today, there is a clear understanding that genetic variants are the main carriers of predictive information on disease pathogenicity and possess two main characteristics:-Penetrance (the percentage of carriers of the corresponding genotype that exhibit the trait);-Expressivity (varying manifestation of the trait in individuals with the same genotype) [98].

This being the case, the terms proposed back in 1925 by Timofeev-Resovsky have turned out to be of such massive importance and far ahead of their time [98], such that today they offer an explanation as to why “mutation” is receding into obscurity as a term, replaced by the term “variant” with its five modifications [97].

Moreover, WES allows for the instantaneous identification of several hereditary diseases in any individual. Reported clinical cases refer to jointly inherited X-linked and autosomal dominant forms of ichthyosis [51], Wilson disease, and hemochromatosis [32,99]. By knowing the molecular defects that lead to the development of the disease, patients may benefit from the most adequate follow-up.

Most diseases are not monogenic; therefore, prior to risk assessment, the nature of a disease shall be properly understood (i.e., monogenic, oligogenic, or multifactorial condition), which is not always easy. The ability to determine persons at a high genetic risk of the most frequent pathologies (diabetes, cardiovascular disease, etc.) at any age presents both opportunities and challenges in clinical medicine. Our studies on hereditary cardiomyopathy [65,66], familial hypercholesterolemia [47], and MODY [13] are an attempt to step up to this challenge. The situation with MFDs is somewhat more complicated, since changes in the genome affect disease etiology, with a large set of genes predisposing to disease (the additivity phenomenon). The disease predisposition is shaped by a large number of environmental factors, while inheritance is not subject to explanation by Mendelian laws only [1]. Indeed, identical diagnoses may be triggered by different risk factors and etiologies in different individuals. Our study shows that WES offers rational algorithms with which to identify genetic markers of complicated diseases, even in limited samples. Genomic medicine can also help identify rare conditions concealed behind a complicated multistep and multicomponent disease diagnosis. Moreover, various common diseases have reported rare genetic variants that increase the MFD risk by several times in heterozygous carriers, e.g., the presence of risk variants for familial hypercholesterolemia in 0.4% of the population, which increase the CAD risk three-fold [100]. Therefore, the risk of monogenic pathologies and MFDs is best assessed via the use of WES.

Molecular medicine and its main areas (predictive medicine, gene therapy, pharmacogenomics, etc.) are to shape a diverse landscape of applied human sciences in the 21st century and potentially the third millennium. Future horizons in genetic testing applications include the following: presymptomatic (pre-emptive) genetic testing (GT) in high-risk families, prospective GT with mandatory follow-up in high-risk individuals based on test findings, and randomized predictive testing [1]. Thus, the concept of predictive medicine—CGPs for solving the problems of preconceptional screening, PGDs, the birth of healthy offspring, diagnostics, and the prevention of MFDs as well as infectious diseases—shall rely on NGS technologies as a fundamental tool using specialized proprietary databases, algorithms, bioinformatics, and genetic custom concepts of expression and penetrance (Figure 5).

Thus, the current advances in PM and their practical value depend on genome sequencing quality and functional analyses within a systems genetics paradigm. The EPMA program suggests a PM roadmap that predominantly relies on the mass sequencing of individual genomes to elucidate their population, ethnic, social, and even interstitial features. The genetic analysis of gene expression allows individual omics profiles to be compared with a patient’s clinical and lab data. Based on such data, integrated gene networks are identified for a patient’s organs and systems most susceptible to pathological processes, and different future developmental scenarios are analyzed. As a result, patients are both the source of information and PM data users [1].

## 4. Conclusions

In conclusion, we would like to emphasize that the scientific foundations of precision medicine and advances in the diagnostics as well as treatment of monogenic, oligogenic, multifactorial, and infectious diseases are driven by the efficient application of NGS technologies jointly with modern analysis algorithms and genetic custom concepts of expression and penetrance. Future horizons to apply genetic testing include the following: presymptomatic (pre-emptive) genetic testing (GT) in high-risk families, prospective GT with mandatory follow-up in high-risk individuals based on test findings, and randomized predictive testing [1]. Knowledge of gene structure, the peculiarities of genetic polymorphisms, and the functions of different variants in the genome with regard to population specificity provide insights into the hereditary nature of a particular monogenic disease or MFD, contributing to their diagnosis, prevention, and treatment, as well as demanding the revision of human diseases’ classification; this, in turn, leads to a paradigm shift and propels the advent and rapid development of molecular medicine as a new science [100].

## Figures and Tables

**Figure 1 jpm-13-01236-f001:**
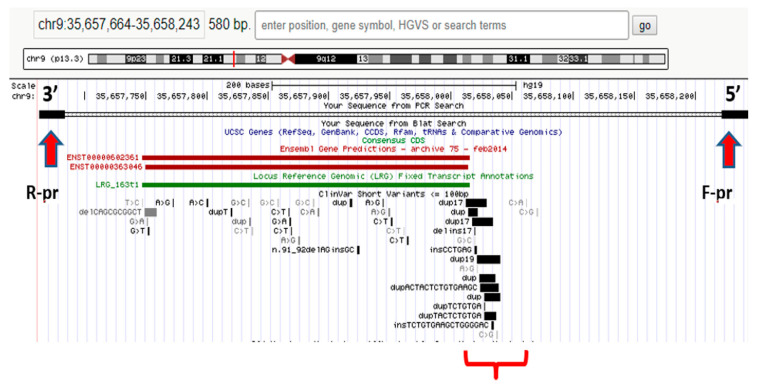
Hotspot *RMRP* gene promoter region in the proband [62].

**Figure 2 jpm-13-01236-f002:**
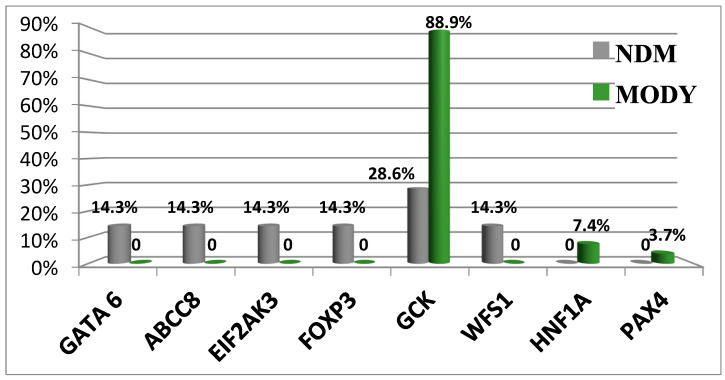
Differences in the prevalence of genetic variants causative of NDM and MODY. Abbreviations: *GATA6*, GATA binding protein 6; *ABCC8*, member 8 of the ATP binding cassette subfamily C; *EIF2AK3*, eukaryotic translation initiation factor 2 alpha kinase 3; *FOXP3*, P3 forkhead box transcriptional factor; *GCK*, glucokinase; *WFS1*, Wolframin ER transmembrane glycoprotein; *HNF1A*, HNF1 homeobox A transcriptional factor; and *PAXA*, paired box A transcriptional factor.

**Figure 3 jpm-13-01236-f003:**
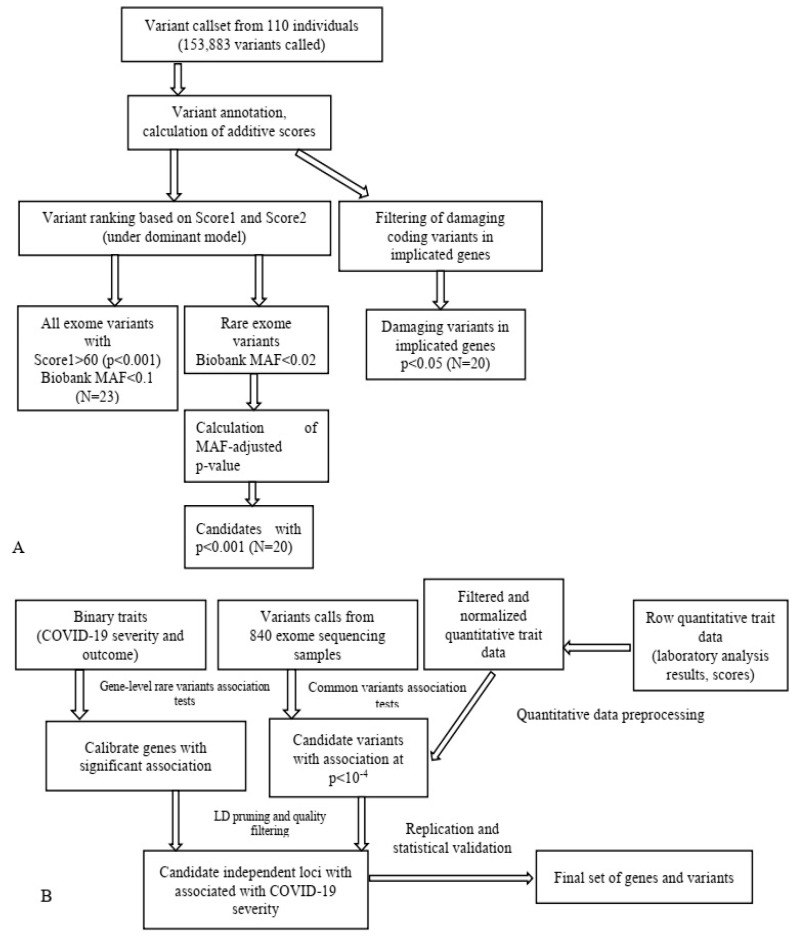
Schematic application of WES data in T2D research (**A**) [16] and data analysis pipeline in a study on COVID-19 (**B**) [14]. MAF, minor allele frequency.

**Figure 4 jpm-13-01236-f004:**
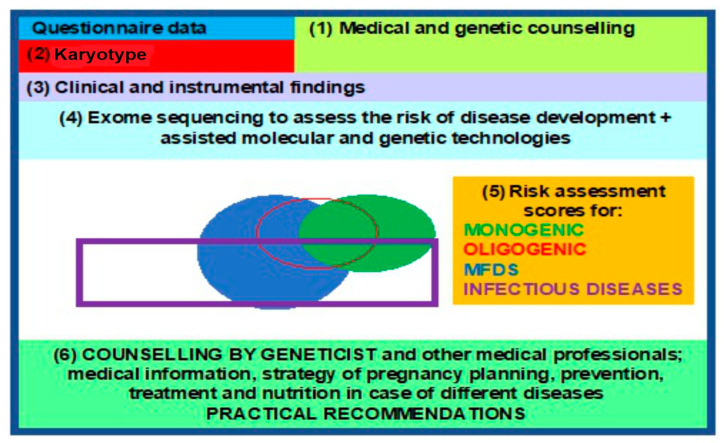
Block diagram for a clinical genetic passport, to aid reproduction.

**Figure 5 jpm-13-01236-f005:**
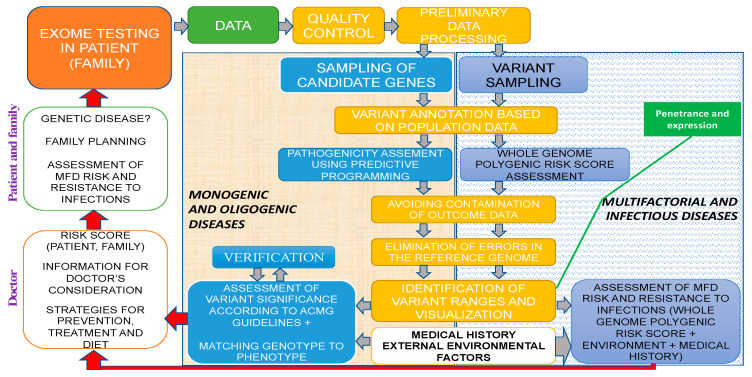
Exome study within the predictive medicine framework of a clinical genetic human health passport. Different colors reflect the sequence of genetic analysis: green for sequencing data, yellow for bioinformatics, gray for the risk analysis of monogenic and oligogenic diseases, and white for MFDs as well as infection diseases.

**Table 1 jpm-13-01236-t001:** Targeted NGS technologies by intended use.

NGS Technology	Study Aims	Working Principle	Data Size, GB	Research and Clinical Applications
WGS	Identification of genetic mutations and SNPs in coding and noncoding genome regions	DNA extraction, fragmentation, sequencing, data analysis, and identification of relevant variants	90–120	Research, diagnosis, and treatment
WES	Identification of variants in protein coding loci and genes (exome)	DNA extraction, fragmentation, target gene identification, sequencing, data analysis, and variant annotation	6–12	Clinical diagnosis, disease-causing genes
TS	Screening for DNA variants affecting numerous genes	DNA extraction, fragmentation, target gene sequence capture, sequencing, data analysis, and variant annotation	0.5–3	Treatment
RNA-seq	Identification of gene expression, new protein isoforms, detection of merging genes, SNPs, insertions, deletions, and small noncoding RNAs	RNA and cDNA extraction, fragmentation, sequencing, data analysis, and variant annotation	3–6	Research and therapy, biomarker discovery, and drug resistance
ChIP-seq		DNA fragmentation, binding beaded antibodies to target proteins, DNA purification, sequencing, and identification of gene variants	1–2	The influence of transcription factors on phenotype-affecting mechanisms and diseases

**Table 2 jpm-13-01236-t002:** MD prevalence in Russia and globally determined by the frequencies of pathogenic SNPs [32].

Disease/Condition	Gene	Allele Count	Carrier Frequency (Lower/Upper CI)	Disease Frequency (Lower/Upper CI)	Known Frequency	References
Retinal dystrophy, Stargardt disease	*ABCA4*	13 (23)	0.0350 (0.0206/0.0589)	3.1 × 10^−4^ (1.1 × 10^−4^/8.8 × 10^−4^)	1 in 10,000 1 in 8000	[33] [34]
Cystic fibrosis	*CFTR*	11 (19)	0.0296 (0.0167/0.0522)	2.2 × 10^−4^ (6.9 × 10^−5^/6.9 × 10^−4^)	1 in 10,000 1 in 3000–16,000	Reported carrier frequency of 0.032 [35] [36]
Phenylketonuria	*PAH*	11 (18)	0.0296 (0.0167/0.0522)	2.2 × 10^−4^ (6.9 × 10^−5^/6.9 × 10^−4^)	1 in 10,000 1 in 4500 [Italy]–1 in 125,000 [Japan]	Reported carrier frequency of 0.029 [37] [38]
Wilson disease	*ATP7B*	4 (6)	0.0108 (0.0042/0.0274)	2.9 × 10^−5^ (4.3 × 10^−6^/1.9 × 10^−4^)	1 in 30,000 1 in 30,000	Similar global incidence reported [39,40]
Galactosemia	*GALT*	4 (5)	0.0108 (0.0042/0.0274)	2.9 × 10^−5^ (4.3 × 10^−6^/1.9 × 10^−4^)	1 in 20,000 1 in 48,000	Reported carrier frequency of 0.006 [35,41]

Abbreviations: ABCA4, ATP binding cassette subfamily A member 4; ATP7B, P-type cation transport ATPase family 7B; CFTR, cystic fibrosis transmembrane regulator; and GALT, galactose-1-phosphate uridylyltransferase.

**Table 3 jpm-13-01236-t003:** Pathogenicity of novel *LDLR* gene variants.

Gene	Patient ID	Exon/Intron	Variant	Allele Frequency in GnomAD	Allele Frequency in [49]	Variant Pathogenicity Classification by ACMG
*LDLR*	G31	4	c.316_328delCCCAAGACGTGCT p.(Lis107Argfs*95)	Not found	Not found	P (PVS1 PS1 PM1 PM2 PP3)
*LDLR*	G29	4	c.325T>G p.(Cys109Gly)	Not found	Not found	LP (PS1 PM1 PM2 PM5 PP3)
*LDLR*	G36	4	c.401G>C (p.Cys134Ser)	Not found	Not found	LP (PS1 PM1 PM2 PM5 PP3)
*LDLR*	1	4	c.433_434insG p(Val145Glyfs*35)	Not found	Not found	P (PVS1 PM2 PP3)
*LDLR*	G18	4	c.616A>C (p.Ser206Arg)	Not found	Not found	Uncertain significance (PM2 PP1 PP3)
*LDLR*	G21	IVS6	c.940+1_c.940+4 delGTGA (g.18154_18157delGTGA)	Not found	Not found	P (PVS1 PM1 PM2 PP3)
*LDLR*	32	8	c.1186G>C p.(Gly396Arg)	Not found	Not found	P (PVS1 PM1 PM2 PM5 PP3)
*LDLR*	G26	IVS8	c.1186+1G>T (g.22279G>T)	Not found	Not found	P (PVS1 PM2 PP3)
*LDLR*	G17	11	c.1684_1691delTGGCCCAA p.(Pro563Hisfs*14)	Not found	Not found	P (PVS1 PM1 PM2 PP3)

Abbreviations: LDLR, low-density lipoprotein receptor; ACMG, American College of Medical Genetics; and P, pathogenic.

**Table 4 jpm-13-01236-t004:** Most efficient diagnostic strategies for hereditary diseases.

Nosology	Efficiency of Diagnostics Prior to NGS, %	Efficiency of Diagnostics after WES, %	Efficiency of Diagnostics with Novel Variants Considered, %	Reference
Cystic fibrosis	45–55 (1 mutation) 58 (35 mutations)	67–80	-	Unpublished
WD	Up to 75 (4 mutations) Up to 86 (12 mutations)	Up to 96	97	[55]
MODY	15–35	40–50	55	[13]
Genetically heterogeneous condition	28%	65%		[56]
Neurometabolic disorder	24%	35%		[56]
Single anomalyy of the fetuses		6%		[57]
Two and more anomalies of the fetuses		35%		[57]
Anomalies of the fetuses		10.3–18.9%		[58]
Anomalies of the fetuses		8.5–15.4%		[59]
Anomalies of the fetuses		6.2–80%		[60]

Abbreviations: MODY, maturity onset diabetes of the young; WD, Wilson’s disease; WES, whole-exome sequencing; and NGS, next-generation sequencing.

**Table 5 jpm-13-01236-t005:** Main HCM genetic variants determined in patients and control group [65].

Gene	Nucleotide Change	Diseased/Risk/Healthy, %	Risk	Risk2	*p*-Value	Polyphen 2	SIFT	Clinical Verification
*MYBPC3*	c.977G>A (NM_000256.3)	5/4/0	19	−99	0.41	Benign	Damaging	[68]
*MYBPC3*	c.2678G>T (NM_000256.3)	5/17/0	16	−96	-	Probably damaging	Damaging	-
*CASQ2*	c.1014+12delG (NM_001232.3)	13/4/0	49	−249	8.62 × 10^−5^	-	-	-
*TNNT2*	c.97+151delC (NM_000364.3)	0/0/10	−100	20	1.80 × 10^−5^	-	-	-
*TNNT2*	c.223+92G>C (NM_000364.3)	0/0/29	−300	60	1.902 × 10^−7^	-	-	-
*TNNT2*	c.223+93C>G (NM_000364.3)	0/0/33	−350	70	2.535 × 10^−4^	-	-	-

Abbreviations: MYBPC3, cardiac myosin binding protein C; CASQ2, Calsequestrin 2; TNNT2, cardiac troponin T; and SIFT, scale-invariant feature transform.

**Table 6 jpm-13-01236-t006:** GPS derivation and testing for five common MFDs [84].

Disease	Discovery GWAS, Case/Control	Prevalence in the Validation Dataset	Prevalence in the Testing Dataset	No. of SNPs in GPS	Tuning Parameter	AUC (95% CI) in the Validation Dataset	AUC (95% CI) in the Testing Dataset
CAD	60,801/123,504	3963/120,280 (3.4%)	8676/288,978 (3.0%)	6,630,150	LDPred (ρ = 0.001)	0.81 (0.80–0.81)	0.81 (0.81–0.81)
Atrial fibrillation	17,931/115,142	2024/120,280 (1.7%)	4576/288,978 (1.6%)	6,730,541	LDPred (ρ = 0.003)	0.77 (0.76–0.78)	0.77 (0.76–0.77)
T2D	26,676/132,532	2785/120,280 (2.4%)	5853/288,978 (2.0%)	6,917,436	LDPred (ρ = 0.01)	0.72 (0.72–0.73)	0.73 (0.72–0.73)
Inflammatory bowel disease	12,882/21,770	1360/120,280 (1.1%)	3102/288,978 (1.1%)	6,907,112	LDPred (ρ = 0.1)	0.63 (0.62–0.65)	0.63 (0.62–0.64)
Breast cancer	122,977/105,974	2576/63,347 (4.1%)	6586/1,576,895 (4.2%)	5218	Pruning and thresholding (r/2 < 0.2; *p* < 5 × 10^−4^	0.68 (0.67–0.69)	0.69 (0.68–0.69)

Abbreviations: CAD, coronary artery disease; T2D, type 2 diabetes; GPS, genome-wide polygenic score; GWAS, genome-wide association study; SNP, single-nucleotide polymorphism; and AUC, area under the ROC Curve.

## Data Availability

The datasets generated during and/or analyzed during the current study are available from the corresponding author upon reasonable request.

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
