# Peer review of "Human Exome Sequencing and Prospects for Predictive Medicine: Analysis of International Data and Own Experience"

_jpm, 2023, doi:10.3390/jpm13081236_

Round 1
Reviewer 1 Report
Manuscript ID Manuscript ID: jpm-2502914
"
Human Exome Sequencing and Prospects for Predictive Medicine. Analysis of international data and own experience " for Journal of personnalized medicine.
Comments to the authors :
- This is a well written manuscript relating the importance of Human Exome Sequencing and its Prospects for Predictive Medicine. This article is doing an Analysis of international data and their own experience.
- There are some modifications that will make the manuscript more relevant and complete:
- Table 1. relating most disease prevalence in Northwest Russia, should be modified and should relate the most genetic disease prevalence in Russia in general not only in Northewest Russia.
- Moreover since this is an international analysis, the authors should add tables relating most genetic diseases worldwide and compare them to stipulate if WES is useful for all populations or not
- Table 3 « Most efficient diagnostic strategies for hereditary diseases ». In this table the authors are relating only some hereditary diseases, not the most frequent, and we don’t understand why they choose only to relate these hereditary diseases. Moreover, fetus with anomalies is not a hereditary disease.
- Figure 1 : why the authors are giving this example and saying the NGS was used to identify mutations of this RMRP gene that is composed only of one exon (264 bp)
- Table 4. Main HCM genetic variants identified in patients and the at-risk group 392 compared to the control group. This table is not well discussed in the manuscript, and the authors should talk more about HCM and all involved genes and not only chose these specific variants.
- Figure 4 : please correct « karyotype » instead of « kariotype ».
- The figure 5 should be modified with less colors and to be more simple to understand for those who will read the article. The figure is not clear and not easy to remember
none
Author Response
Dear reviewer,
Thanks very much for your comments on our manuscript.
1.We have changed table 1, added the prevalence of diseases throughout Russia and in the world, added new links to these diseases. (we have added a new table 1 (as noted by another reviewer) so table 1 is now number 2.)
2. In Table 3, we have corrected fetuses with abnormalities, for fetal abnormalities. We chose these diseases because they were our research in Russia. (Please see you table 4).
3. We show RMRP gene in Figure 1 because we performed WES and found a new variant in this gene in a patient with skeletal chondrodysplasia (anauxetic dysplasia)
4. To table 4 (now it is table 5) we have added a more detailed description of Hypertrophic cardiomyopathy
5. In figure 4, we corrected the word karyotype
6. We have uploaded a new drawing 5 in the best quality. To figure 5, we gave explanations regarding the colors in the figure. Different colors reflect the sequence of genetic analysis: obtaining sequencing data a green color, bioinformatics – a yellow color, risk analysis of monogenic and oligogenic diseases – a gray color, MFDs and infection disease – a white color.
See our corrections in the manuscript

Reviewer 2 Report
This review discusses the current use of whole exome sequencing (WES) for screening genetic changes associated with diseases. It emphasizes the potential of WES not only for diagnosing monogenic diseases but also for assessing risks and understanding a wide range of conditions in healthy individuals. The review presents a new clinical concept of the human genetic passport, utilizing WES to identify candidate genes, variants, and molecular mechanisms for diagnosis, prediction, and treatment of monogenic, oligogenic, and multifactorial diseases. The review discusses the various possibilities of WES using examples from different diseases and explores the advantages and disadvantages of this sequencing method. Overall, this review provides a good summary with comprehensive analyses of international data along with the authors’ own experience. I have no major comments for the manuscript. Please find my minor comments to improve the manuscript as follows:
1. Consider including a diagram, graph, or table summarizing the advantages and limitations of WES. This visual representation will help readers grasp the key points more effectively.
2. Authors may discuss potential solutions for unsolved cases and highlight the fact that no sequencing method can achieve a 100% diagnostic rate. Consider mentioning options such as seeking assistance from undiagnosed disease networks or collaborating with researchers to further characterize variants of uncertain significance.
3. Pay attention to abbreviated words and ensure that full names are provided where necessary. This will enhance clarity and avoid confusion for readers.
Conduct a thorough check for English and spelling errors to ensure the manuscript's overall quality.
Author Response
Dear reviewer,
Thanks very much for your comments on our manuscript.
1. We have included a table (number 1) in the introduction summarizing the advantages and disadvantages of WES.
2.We included in the article that WES cannot achieve a 100% diagnostic rate. Therefore, will need seeking assistance from undiagnosed disease networks or collaborating with researchers to further char-acterize VUS variants.
3. We checked the abbreviations and the name of the genes.
4. We conducted a thorough check for English and spelling errors in the manuscript.
See our corrections in the manuscript

Reviewer 3 Report
In this paper, the authors discussed the advantages and disadvantages of the whole exome sequencing. Moreover they demonstrate the different possibilities of WES using the example of various diseases. The article presented by the authors is interesting; however, they do not add anything new.
The paper is excessively long; in all sections, some paragraphs must be eliminated or reduced. For example, in the introduction, the paragraph in line 84 could be reduced, in the same way as the paragraph of line 101.
The authors have widely explained all tables in the text. The tables are introduced to reduce the text in the article and not increase it, as in this case. The authors must reduce the extension of these paragraphs.
Moreover, the content of sections 2.2 and 2.3 must also be reduced or structured in a different way. The results of other authors are explained widely. The authors must also rewrite these paragraphs.
The conclusions are also too extensive. This section should collect the most relevant conclusions or information for the article and the figures should not be included in this section.
Author Response
Dear reviewer,
Thanks very much for your comments on our manuscript.
1. We reduced paragraphs in the line of 84, 101 and others
2.We reduced the paragraphs which present in the tables 2, 3, 4, 5, 6 in the text. 3. We reduced and rewrited the content of 2.2 and 2.3 paragraphs.
4. We reduced the conclusions. Also we added the section of 2.5. Clinical Genetic Passport (CGP)
See our corrections in the manuscript
